A survey of the sperm whale (Physeter catodon) commensal microbiome

Li Chang 1 2
Tan Xiaoxuan 2
Bai Jie 2
Xu Qiwu 2
Liu Shanshan 2 3
Guo Wenjie 2
Yu Cong 2
Fan Guangyi 2 4 5
Lu Yishan 6
Zhang He 2
Yang Huanming 1 5 7
Chen Jianwei chenjianwei@genomics.cn 2
Liu Xin liuxin@genomics.cn 2 5 7 8
1 BGI Education Center, University of Chinese Academy of Sciences , Shenzhen , China
2 BGI-Qingdao, BGI-Shenzhen , Qingdao , China
3 Shandong Technology Innovation Center of Synthetic Biology , Qingdao , China
4 State Key Laboratory of Quality Research of Chinese Medicine and Institute of Chinese Medical Sciences, University of Macau , Macau , China
5 BGI-Shenzhen , Shenzhen , China
6 Guangdong Ocean University , Shenzhen , China
7 China National GeneBank, BGI-Shenzhen , Shenzhen , China
8 State Key Laboratory of Agricultural Genomics , Shenzhen , China
Rappe Michael
Electronic publication date: 2019 Jul 4
Publication date: 2019
Volume: 7
Electronic Location ID: e7257
Received 2018 Dec 18; Accepted 2019 Jun 5
Copyright: ©2019 Li et al.
Copyright year: 2019
Copyright holder: Li et al.
License: This is an open access article distributed under the terms of the Creative Commons Attribution License, which permits unrestricted use, distribution, reproduction and adaptation in any medium and for any purpose provided that it is properly attributed. For attribution, the original author(s), title, publication source (PeerJ) and either DOI or URL of the article must be cited.
License URL: https://creativecommons.org/licenses/by/4.0/

Keywords: Sperm whale, Commensal microbiome, Pathogentic microorganisms

Funding: The National Key Research and Development Program of China 2016YFE0122000 Shenzhen Peacock Plan KQTD20150330171505310 The National Key Research and Development Program of China (No. 2016YFE0122000) and Shenzhen Peacock Plan (No. KQTD20150330171505310) supported this study. There are no additional extenal funding received from this study. The funders had no role in study design, data collection and analysis, decision to publish, or preparation of the manuscript.

==============================
Background

Mammalian commensal microbiota play important roles in the health of its host. In comparison to terrestrial mammals, commensal microbiota of marine mammals is mainly focused on the composition and function of skin and gut microbiota, with less attention paid to the health impact of bacteria and viruses. Previous studies on sperm whales (Physeter catodon) have affirmed their important phylogenetic position; however, studies on their commensal microbiota have not been published, due to difficulty in sample collection.

Methods

Here, we sequenced the metagenomes of blood, muscle and fecal samples from a stranded sperm whale using the BGISEQ-500 platform. We compared the diversity and abundance of microbiomes from three different tissues and tried to search pathogenic bacterial and virulence genes probably related to the health of the sperm whale. We also performed 16S rDNA sequencing of the fecal sample to compare to published gut metagenome data from other marine mammals.

Results

Our results demonstrated notable differences in species richness and abundance in the three samples. Extensive bacteria, including Enterococcus faecium, Fusobacterium nucleatum, Pseudomonas aeruginosa, Streptococcus anginosus, Streptococcus pneumoniae, and Streptococcus suis, and five toxigenic Clostridium species usually associated with infection, were found in the three samples. We also found the taxa composition of sperm whale gut microbiota was similar to that of other whales, suggesting co-evolution with its host. This study is the first report of the sperm whale gut microbiome, and provides a foundation for the pathogen detection and health assessment of the sperm whale.

Introduction

Microorganisms exist widely in mammal bodies and the environment, and they are indispensable to the health of mammals (Cho & Blaser, 2012; Human Microbiome Project, 2012; McFall-Ngai, Hadfield & Bosch, 2013). Marine mammals are emblematic members of aquatic ecosystems, but marine mammal populations are declining gradually, and may be progressively impacted by climate change, environmental pollution, and direct (e.g., hunting) and indirect (e.g., habitat exploitation) anthropogenic activities (Moore, 2008; Pompa, Ehrlich & Ceballos, 2011). There is growing concern about the current health status and life habitat of marine mammals. The study of marine mammalian symbiotic microbes is important to understand the health of marine mammals and their conservation.

Until now, investigations of marine mammalian symbiotic microbes were mainly focused on the composition and function of skin microbiota communities in order to examine the potential for a core bacterial community and its variability with specific host or environmental factors (Apprill, Robbins & Eren, 2014; Hooper & Brealey, 2019; Bierlich et al., 2018). The composition of the gut microbiomes of marine mammals have also been extensively studied, focusing on the putative functionality of these symbiotic communities, differences between marine mammals, and correlationship between marine mammals’ diet and evolution (Bik, Costello & Switzer, 2016; Erwin et al., 2017; Merson, Ouwerkerk & Gulino, 2014; Nelson et al., 2013; Sanders et al., 2015). As far as diseases in marine mammals have been concerned, few studies have been conducted on the health impact of bacteria and viruses on marine mammals, which might be limited due to the difficulties in sample collection and the opportunistic nature of stranding events (Godoy-Vitorino, Rodriguez-Hilario & Alves, 2017; Van Bressem et al., 2014). More marine mammals’ samples are required to confirm these studies and apply this knowledge to conservation and rehabilitation.

The sperm whale (Physeter catodon) is the largest toothed whale and the only living member of genus Physeter (Marino, 2004). As a representative species for Physeteridae, Odontoceti and Cetacea, they have unique adaptations and a phylogenetically important position (Warren, Kuderna & Alexander, 2018). They feed on several species, most notably the squid, but also octopuses, and fish such as demersal rays (Smith & Whitehead, 2000). Recently, the chromosome level genome sequence of the sperm whale has been published (Fan et al., 2019), providing a good genetic foundation for studying its conservation and population structure. However, the commensal microbiome of sperm whale based on metagenomic sequencing has not been reported yet. The diversity, composition and structure of sperm whale symbiotic microbiota are not known.

Hence, a survey of the sperm whale commensal microbiome is very important for understanding its specific microorganisms and finding some possible pathogenic microorganisms related to its health. In this study, we collected a female stranded sperm whale and characterized the blood, muscle and fecal microbiome using metagenomic sequencing. We also screened for pathogenic microorganisms and virulence genes which might be related to its health in these samples. Additionally, we investigated its gut microbiomes using 16S community sequence analysis to detect the unique microbiome taxa composition by comparing it to published cetacean microbiomes. Our reference metagenome data and the potential pathogenic microbes could be used to further monitor and evaluate the health of sperm whales and other marine mammals.

Materials and Methods

Sample collection and DNA extraction

The present study sampled a living female sperm whale stranded near the bay area of Huizhou City in southern China on 12th March 2017, and we collected a blood sample of about 5 ml from the vein via syringe (Fig. S1). The whale died on 15th March, approximately 79 h after rescue. The dead animal was salvaged and post-mortem analysis was performed according to standard protocols (Kuiken, 1991). We collected fecal and muscle samples during the necropsy. We dissected the intestine with sterile scissors and forceps, then we used a swab to collect about 2 g of fecal matter from inside the incision to a sterile plastic tube. To collect a muscle sample, we cut the sperm whale cortex tissue and about 5g of muscle tissue was collected with scissors. Each sample was immediately stored in −20 °C freezers before transport to the laboratory within 12 h for further investigation.

The blood sample was combined with RNAlater to dissolve eukaryotic cells, and then centrifuged to obtain bacteria cells to extract DNA. Fecal and muscle sample DNA extractions were performed using the QIAamp DNA Stool Mini Kit (Qiagen, Valencia, CA, USA). To eliminate RNA contamination, DNase-free RNase was used to treat extracts. Finally, DNA quantity was determined using a Qubit 3.0 fluorometer, and DNA integrity was evaluated by gel electrophoresis (Pan et al., 2018). All lab work in this study was conducted in a sterile flow hood to prevent contamination. Blood and other samples were dealt in different laboratories to prevent same microbiota contamination. This study was approved by the Institutional Review Board of BGI (NO. BGI-R052-3 and NO.FT 17160) and the Institute of Deep-sea Science and Engineering Chinese Academy of Sciences (NO.SIDSSE-SYLL-MMMBL-01).

Library construction and sequencing

We totally constructed three metagenome pair-end libraries for the three samples and one 16S rDNA amplicon library for the fecal sample. To construct metagenome sequencing libraries, the extracted DNA was sheared into fragments between 500 bp and ∼800 bp in size. Fragments between 150 bp and 250 bp were selected using AMPure XP beads (Agencourt, Beverly, MA, USA) and then repaired using T4 DNA polymerase (ENZYMATICS, Beverly, MA, USA) to obtain blunt ends which were then 3-adenlyated to create sticky ends. These DNA fragments were ligated at both ends to T-tailed adapters and amplified for eight cycles. These amplification products were subjected to a single-strand circularization process using T4 DNA Ligase to generate a single-stranded circular DNA library. For constructing 16S V4 rDNA sequencing library, binding sites for sequencing primers of BGISEQ-500 were added at the 5′ end of V4 region common primers (515F: GTGCCAGCMGCCGCGGTAA, 806R: GGACTACHVGGGTWTCTAAT). The PCR products with target bands were mixed in equal amounts, and 2% agarose gel was used for electrophoresis and gel cutting, and then were subjected to generate a single-stranded circular DNA library.

All libraries were sequenced on the BGISEQ-500 platform in the paired-end model with 100 bp length reads for metagenome in BGI-Shenzhen and 150 bp for 16S rDNA library in BGI-Qingdao. Different sequencing machines were used for different libraries to prevent same microbiota contamination.

Metagenome assembly and taxonomic assignment

Adaptor contamination, low quality reads and duplication for metagenomes were filtered from the raw reads by SOAPnuke (1.5.6) (Chen, Chen & Shi, 2018), and the metagenome remaining reads were removed to eliminate host DNA on the basis of sperm whale reference (Fan et al., 2019) using SOAP2 (Version 2.21) (Gu, Fang & Xu, 2013) with the parameters “-v 8 -m 4”. To account for the possibility of contamination with human-associated microorganisms, clean data were mapped to the human genome, and samtools (Version 1.9) with the parameters “view -q 30 -F 4 -F 256” was used to account mapping reads. After removing the host data, the unmapped metagenome data of 16 Gb, 53 Gb and 15 Gb data for the blood, fecal and muscle samples, respectively, were subjected to further analysis. We then used metaSPAdes (Version 3.10.1) (Nurk, Bankevich & Antipov, 2013) to assemble each sample separately. MetaGeneMark (Version 3.38) (Zhu, Lomsadze & Borodovsky, 2010) was used to identify the coding sequences (CDSs) of the assembled sequences.

Taxonomic assignment of the predicted genes was carried out using BLAST+ (Version 2.2.26, best match according to BlastN, evalue <1e−5, coverage >50%) by aligning against the prokaryotic representative genomes (Version 201608) and non-redundant nucleotide (NT) database (Version 20170616) (Zhao et al., 2018). The reads were then aligned to the predicted genes using Bowtie2 (Version 2.2.5) (Langmead & Salzberg, 2012), thus the sequence-based gene abundance profiling was calculated using PathoScope (Version 2.0.6) (Francis et al., 2013). The relative abundances of species were calculated from the relative abundance of their respective genes, and the microbes with a relative abundance below 1e−8 were removed to reduce contamination (Guo et al., 2018).

The blood, muscle and fecal clean reads were aligned to the VFDB (Virulence Factors of Bacterial Pathogens) database using Bowtie2 (Version 2.2.5) and the blood assembly contigs were aligned using blat (Version 3.2.1). The Virulence Factors identified in the two results that were annotated to Clostridium novyi and Clostridium botulinum were selected. The core bacterial virulent genes were aligned by MUSCLE (Version 3.8.31) with the same bacteria genome from NCBI and the trees were built by FastTree (Price, Dehal & Arkin, 2009).

16S rDNA community sequence analysis

In order to compare the gut microbiota with that of other marine mammals, we first collected the 16S V4 rDNA tags of the fecal samples of six species from the NCBI and EMBI database (see more data details in Table S1). Then the tags were clustered to the OTUs (Operational Taxonomic Units) using USEARCH (Version 7.0.1090) (Rognes et al., 2016) with a 97% threshold. Taxonomic assignment of OTU sequences were classified using Ribosomal Database Project (RDP) Classifier (v.2.2) trained on the RDP database (version trainset14_032015) (Cole et al., 2014), using 0.8 confidence values as a cutoff. The data of sperm whale fecal sample 16S rDNA amplicon was discarded to eliminate the low quality and adapter pollution, then paired-end reads with overlap were merged to tags using FLASH (v1.2.11). Finally, all tags were mapped into the OTU representative sequences using USEARCH GLOBAL to obtain the OTUs and species abundance profiles. Bray-Curtis distance was used to estimate β-diversity (between-sample diversity).

Results

Community diversity and structure of the samples’ microbiome

To obtain a reference metagenomic sequences of the three tissues, we performed a strict data quality control and obtained ∼16 Gb, ∼53 Gb and ∼15 Gb high-quality data of blood, fecal and muscle samples, respectively (Table S2). Subsequently, we assembled the metagenomic reads of the three tissues, with an assembly size of ∼37 Mb, ∼470 Mb and ∼151 Mb for blood sample, fecal sample and muscle sample, respectively (Table 1). A total of 1,715 microorganisms were found in the three tissues, among which the microbial species in the fecal samples were the most abundant. There were notable differences in species richness and abundance in the different samples (Fig. 1A).

Table 1 Statistics of metagenomics assembly for three tissues.

Statistics	Blood	Fecal	Muscle	
	Contig	CDS	Contig	CDS	Contig	CDS	
Total number	40,086	61,992	287,646	628,309	155,131	174,718	
Total length(Mb)	37.16	27.83	470.47	375.63	151.50	43.71	
N50 length (bp)	1,930	717	5,153	963	2,060	306	
N90 length (bp)	309	213	498	294	314	123	
Max length (bp)	488,134	10,443	546,844	29,826	112,716	147,900	
Min length (bp)	200	57	300	57	200	57	
GC content	32.73%	32.94%	48.30%	49.18%	49.97%	52.26%	
Mapped reads	91.43%	66.33%	88.87%	57.86%	18.57%	7.11%	

Figure 1 Host-associated microbial communities of sperm whale.

(A) A PCA of species with sperm whale blood, fecal and muscle samples. (B) The taxonomic distribution of microbes detected at least in two tissues by metagenomics analysis of sperm whale blood, fecal and muscle samples. The relative abundance genera from the top 20 abundant genera are shown.

In all three tissues, only 113 microbial species were detectable, and they were those belonging to seven phyla Firmicutes, Bacteroidetes, Deferribacteres, Spirochaetes, Fusobacteria, Proteobacteria and Tenericutes (Table S3). A total of 21 phyla and 74 genera were identified in at least two tissues (Fig. 1B). Each sample had a dominance of different taxa; for instance, the blood sample was dominated by Firmicutes. In comparison with blood and muscle samples, the fecal sample was more abundant in Bacteroidetes and Firmicutes. The muscle sample shared fewer genera when compared to the other two samples.

Detection of potentially pathogenic microorganisms

To detect potentially pathogenic microbes of the sperm whale, we screened the pathogenic bacterial species in the metagenome sequence data of three tissues. We found pathogenic microbes including Enterococcus faecium, Fusobacterium nucleatum, Pseudomonas aeruginosa, Streptococcus anginosus, Streptococcus pneumoniae, Streptococcus suis, and five toxigenic Clostridium in the bloodstream and intestine, and six of them appeared in muscle tissue (Table 2) (Chevalier, Bouffartigues & Bodilis, 2017; Engholm, Kilian & Goodsell, 2017; Fortier, 2017; Gao, Howden & Stinear, 2018; Han, 2015; Okwumabua et al., 2017). Then we aligned the blood, muscle and fecal assembly results and sequencing data against the VFDB databases to identify bacterial virulence genes (Chen et al., 2016). Twelve virulence genes were found in Clostridium novyi and Clostridium botulinum. By aligning the pathogenic genes to all strain sequences of C. novyi and C. botulinum, we found the nearest bacteria to them were C. novyi A str. 4540 (GCA_000724445) and C. botulinum C str. Stockholm (GCA_000219255) from Sweden (Figs. 2A and 2B).

Table 2 Detection of pathogenic bacteria and nematodes in fecal, blood and muscle samples.

Species	Blood	Fecal	Muscle	
Clostridium baratii	+	+	+	
Clostridium botulinum	+	+	+	
Clostridium novyi	+	+	+	
Clostridium perfringens	+	+	+	
Clostridium tetani	+	+	+	
Enterococcus faecium	+	+	–	
Fusobacterium nucleatum	+	+	+	
Pseudomonas aeruginosa	+	+	–	
Streptococcus anginosus	+	+	–	
Streptococcus pneumoniae	+	+	–	
Streptococcus suis	+	+	–	
Bacteroides fragilis	–	+	–	
Enterococcus faecalis	–	+	–	

Figure 2 Phylogeny tree of all strain sequences virulence genes found in C. botulinum and C. novyi.

(A) Phylogeny tree of four virulence genes found in C. botulinum. All C. botulinum strains genome from NCBI were shown. (B) Phylogeny tree of nine virulence genes found in C. novyi. All C. novyi strains genome from NCBI were shown.

Comparison of gut microbiome in different marine mammals

The composition of mammalian gut microbiota could be shaped by host diet, age and phylogenetic position (Hullar & Fu, 2014; Muegge, Kuczynski & Knights, 2011). To profile the unique gut microbiota of sperm whales, we compared the composition of gut microbiome between sperm whales and other marine mammals. A total of 122 bacterial Operational Taxonomic Units (OTUs) were identified in the sperm whale’s gut microbiota (Table S4). The majority of microbes were members of the phyla Bacteroidetes (51.49% of the total) and Firmicutes (31.18% of the total), as well as the phyla Euryarchaeota, Spirochaetes and Verrucomicrobia (Fig. 3A). The sperm whale’s sample had either very few or no reads assigned to Proteobacteria, which was comparatively common among six other marine mammals.

Figure 3 Gut microbiomes of sperm whale fecal sample.

(A) Phylum composition distribution of sperm whale and six marine mammals’ gut microbiomes. Beluga whale (Delphinapterus leucas) and dolphin (Tursiops truncatus) are tooth whale. Humpback whale (Megaptera novaeangliae), sei whale (Balaenoptera borealis), and right whale (Eubalaena glacialis) are baleen whale. Manatee (Florida manatee) is belongs to Sirenia. (B) Phylogenetic tree basic on Bray–Curtis distance in mammalian gut microbiomes.

We also calculated these compositional dissimilarities using β-diversity (Whittaker, 1960), and built cluster trees using β-diversity to disentangle the effect of different factors shaping community assembly at different phylogenetic scales. The Bray–Curtis distance in mammalian gut microbiomes showed that the sperm whale was closest to the Beluga Whale (Delphinapterus leucas) and furthest from the manatee (Florida manatee), which is consistent with the biological evolutionary tree of these marine species (Fig. 3B) (Groussin et al., 2017).

Discussion

Here, the first microbiome inventory of a stranded sperm whale was presented. Among previous metagenomic studies, few have focused on the commensal microbiome of stranded marine mammals, possibly due to the high contamination risk during sample collection (Godoy-Vitorino, Rodriguez-Hilario & Alves, 2017). In this study, we significantly focused on sperm whale salvage in order to ensure that in the collection of fecal and muscle samples, there was no contamination by the external environment. To monitor DNA extraction kits and other laboratory reagent contamination (Lusk, 2014; Salter et al., 2014), we set up a negative control for the meta 16S rDNA library (Fig. S2) and removed microbes with relative abundance below 1e−8 for metagenomic data (Guo et al., 2018). To reduce possibility contamination with human-associated microorganisms, we removed the reads which can map to human genome with high quality (Hooper & Brealey, 2019). The commensal microbiome of the stranded sperm whale in this study contained five main phyla (>1% relative abundance), which were also reported in pygmy (Kogia breviceps) and dwarf (K. sima) sperm whales, corroborating the validity of contamination control in this study (Erwin et al., 2017; Godoy-Vitorino, Rodriguez-Hilario & Alves, 2017).

The pathogens detected in this study, including E. faecium, P. aeruginosa, S. anginosus, S. pneumoniae, S. suis, and F. nucleatum, are opportunistic pathogens and can cause blood infection and immunity damage (Asam & Spellerberg, 2014; Engholm, Kilian & Goodsell, 2017; Gao, Howden & Stinear, 2018; Mulcahy, Isabella & Lewis, 2014; Prithiviraj, Bais & Weir, 2005). F. nucleatum and five toxigenic Clostridium were found in all three samples, and F. nucleatum is associated with a wide spectrum of human diseases (Bashir, Miskeen & Hazari, 2016; Han, 2015; Signat et al., 2011). Twelve virulence genes were found in C. novyi and C. botulinum which were the nearest two highly virulent bacteria of C. novyi A str. 4540 (GCA_000724445) and C. botulinum C str. Stockholm (GCA_000219255) from Sweden (Figs. 2A and 2B). C. novyi A can cause Black disease and is characterized as lethal and necrotizing (Kahn, 2005; Skarin & Segerman, 2014), and C. botulinum C (group III) can cause diseases in animals. This suggests that C. novyi and C. botulinum found in this study may also affect host health (Skarin et al., 2011).

Many pathogenic bacteria have been found in humans and other economic animals (Baumler & Sperandio, 2016), although the reports of them in marine mammals were limited. Live stranding causes a number of Cetacean deaths (Kemper et al., 2005; Evans et al., 2002), and bacterial infection has been suggested to be an important and underestimated factor leading to stranding of marine mammals (Cools, Haelters & Lopes dos Santos Santiago, 2013; Cowan, House & House, 2001). The pathogens we detected also can be used to monitor the health of marine mammals. Additionally, we found the pathogenic nematodes Elaeophora elaphi in the metagenome data of blood that was supported by our dissection result (Fig. S3) (Hernandez Rodriguez & Martinez Gomez, 1986). This also proved that this is a comprehensive method to understand the overall health of the marine mammals.

Additionally, the order of pathogenic species found in fecal samples were also identified in meta 16S rDNA data analysis results. Genomic DNA was extracted from the blood sample and other samples from different laboratories, and metagenomic library construction and meta 16S library construction were also independent. Thus, the pathogenic microorganisms in the three samples were real components of the sperm whale commercial microbiome (Hooper & Brealey, 2019).

Moreover, when comparing the gut microbes of other marine mammals, we performed 16S V4 sequencing for the sperm whale fecal sample instead of using metagenomics data. Since most of the published gut microbe data of marine mammals are 16S data, we maintained consistency with them to ensure the accuracy of the analysis results (Sanders et al., 2015). High abundance of Bacteroidetes and Firmicutes identified in the gut of the sperm whale were similar with pygmy and dwarf sperm whales, indicating similar richness and evenness of gut microbiomes in these closely related species (Erwin et al., 2017; Langer, 2001; Sanders et al., 2015; Zhu et al., 2011). Interestingly, the β-diversity phylogenetic tree is consistent with the evolution of host, suggesting that sperm whales and their gut microbiota have a coevolutionary relationship that was also consistent with reports from previous research (Groussin et al., 2017; Sanders et al., 2015). In the future, more gut microbiome data of sperm whales and other marine mammals should be studied to identify the structural determinants of gut microbiome composition in marine mammals.

Conclusions

As the first metagenomic survey of sperm whale blood, fecal and muscle microbiome using next-generation sequencing, our results showed the differences in symbiotic microorganisms among the three tissues, and demonstrated the gut microbiota of sperm whales has a coevolutionary relationship with its host. The pathogens we detected in the blood, muscle and fecal samples may cause some health problems in the sperm whale, knowledge which will contribute to the monitoring of the health of marine mammals.

Supplemental Information

Table S1 Information of marine mammals’ gut microbiome data

Click here for additional data file.

Table S2 Summary of metagenomics sequence data for sperm whale

Click here for additional data file.

Table S3 The abundance of the microbial species which were detectable in all three tissues

Phyla level appears in this table.

Click here for additional data file.

Table S4 Gut microbiomes OTU table of sperm whale and other marine mammals

Click here for additional data file.

Figure S1 An overview of the sequencing and taxonomic assignment workflow

Click here for additional data file.

Figure S2 Electrophoresis gel figure of negative control for the 16S library

M is the marker, B is the negative control, S is 16S V4 library.

Click here for additional data file.

Figure S3 The parasites in our sequencing individual

Click here for additional data file.

We thank Guangdong Ocean University and the Institute of Deep-sea Science and Engineering Chinese Academy of Sciences for collecting samples. We thank Xiangqun Chi, Santasree Banerjee and Zhangyi Liu from BGI-Qingdao for revising the manuscript. We thank Shandong Technology Innovation Center of Synthetic Biology for their advises to support this work.

Additional Information and Declarations

Competing Interests

Author Contributions

Animal Ethics

DNA Deposition

Data Availability

The authors declare there are no competing interests.

Chang Li and Jianwei Chen conceived and designed the experiments, analyzed the data, prepared figures and/or tables, authored or reviewed drafts of the paper, approved the final draft.

Xiaoxuan Tan analyzed the data, prepared figures and/or tables.

Jie Bai contributed reagents/materials/analysis tools, approved the final draft.

Qiwu Xu, Wenjie Guo and Cong Yu performed the experiments.

Shanshan Liu, Yishan Lu and He Zhang contributed reagents/materials/analysis tools.

Guangyi Fan conceived and designed the experiments, authored or reviewed drafts of the paper, approved the final draft.

Huanming Yang authored or reviewed drafts of the paper.

Xin Liu conceived and designed the experiments.

The following information was supplied relating to ethical approvals (i.e., approving body and any reference numbers):

The Institutional Review Board of BGI (NO. BGI-R052-3 and NO.FT 17160) and the Institute of Deep-sea Science and Engineering Chinese Academy of Sciences (SIDSSE-SYLL-MMMBL-01) provided full approval for this research. Approval for sampling of blood from the live whale is covered by approval NO.FT 17160 and SIDSSE-SYLL-MMMBL-01.

The following information was supplied regarding the deposition of DNA sequences:

The DNA sequencing data and genome assembly are available at the NCBI Sequence Read Archive (PRJNA411766) and GenBank (CNP0000020).

The following information was supplied regarding data availability:

The raw data is available at the NCBI Sequence Read Archive and China National GeneBank: PRJNA411766 and CNP0000020 (https://db.cngb.org/search/project/CNP0000020/) respectively. The accession number of fecal sample metagenomics data is SRR6192929. The data accession number of blood sample metagenomics data is SRR6192931. The data accession number of muscle sample metagenomics data is SRR6192927 and SRR6192928. The China National GeneBank accession number of fecal sample meta 16S rDNA data is CNR0065492 (https://db.cngb.org/search/run/CNR0065492/).

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
