# Peer review of "A survey of the sperm whale (Physeter catodon) commensal microbiome"

_PeerJ, doi:10.7717/peerj.7257_

## Round 0.1 · original submission · Major Revisions

It is important for the authors to understand that I was conflicted about whether to recommend major revision or rejection, as major aspects of this work need to be carefully reconsidered. Many of the authors' conclusions are not supported by their findings. This manuscript requires a significant re-write prior to being acceptable for publication.

Reviewer 1 ·

Basic reporting

- The manuscript needs to be proof-read and edited to ensure correct language and grammar are used throughout. Currently, many sentences do not make sense and the meaning is often ambiguous.

- The introduction does not clearly lay out the aim of the study. The authors suggest that not much work on cetacean microbiomes has been undertaken, but there is a moderate amount of literature surrounding cetacean microbiomes (e.g. see Apprill et al 2014, Humpback whales share a core skin bacterial community: towards a health index for marine mammals?; Bierlich et al, 2018, Temporal and regional variability in the skin microbiome of humpback whales along the Western Antarctic peninsula; Sanders et al, 2015, Baleen whales host a unique gut microbiome with similarities to both carnivores and herbovires, etc) , albeit not of stranded individuals (although see Godoy-Vitorino et al, 2017, The microbiome of a striped dolphin stranded in Portugal) . The authors go on to suggest that stranding is a large issue in cetacean biology (I suppose specifically in conservation, although this isn’t mentioned and perhaps should be), and that this research will shed light on the microbiome related to host physiological characteristics, deep-sea adaptation and stranding. This creates an unclear perspective of what this paper is about, particularly because neither physiological characteristics nor deep-sea adaptations are further discussed. I would suggest re-writing the introduction with a focus on what is currently known about the cetacean microbiome (citing papers which have explored this already), and that this study addresses the gap in knowledge we have about the microbiome of stranded cetaceans.

- The discussion is too short and does not actually discuss the findings of the paper in depth. The authors should focus more on what their results actually show (at a fine-scale) and how this links with the broader literature of cetacean microbiomes (and microbiome work in general).

- I could not find the raw data. The NCBI accession number shared by the authors took me to the sperm whale genome, but not to the data generated in this study (that I could see)

- Figures and tables:

Figure 1A: A PCA of species with sample type overlaid would be a clearer and more informative way to visualise this data. If a Venn diagram must be used, it would be helpful to at least have the circle sizes proportional to the number of species found.

Figure 1B: clarification needed. Is this the top 20 most abundance species across all tissue types?

Figure 2A: the proportions of firmicutes and euaryarchaeota don’t seem to match the proportions reported in line 158 of the text

Table 4 is not mentioned in the text

Figure 3C is missing

Figure 3B: it would be good to have species names here (could still be coloured by clade e.g. dolphin/whale).

Experimental design

- This is an opportunistic study which took advantage of a rare event occurring - the stranding of a sperm whale with the opportunity to sample it - and the results, although interesting and providing new knowledge to the field of microbiome research, must be seen within this context. In particular, it must be acknowledged that there is a sample size of 1 in this study, and therefore the scope of what it can tell us about the sperm whale microbiome is extremely limited.

- The research question is not well-defined by the authors. As previously mentioned, the introduction needs to be rewritten in order to put this study in context, and it must also clearly define the research question. Arguably, the aim of the study is to characterise the blood, muscle and fecal microbiome of a stranded sperm whale, compare the results with published cetacean microbiomes to understand whether the microbiome of a stranded sample still shows similarity to other cetacean samples, and explore *possible* pathogenic bacterial species and parasitic species present in the whale.

- Comments on the methods:

My largest concern in the methods is the lack of control for, or discussion of, contamination.

Contamination is a common problem in microbiome studies (e.g. see Salter SJ, Cox MJ, Turek EM, Calus ST, Cookson WO, Moffatt MF, et al. Reagent and laboratory contamination can critically impact sequence-based microbiome analyses. BMC Biol. 2014;12:87.), and can be at least partially controlled with the inclusion of blanks at the sequencing stage. There is no discussion of blanks in this paper, and I therefore assume that they were not included. As a late-stage method to understand whether contamination from human skin may influence results, the authors could align the unfiltered data to the human genome to understand whether human contamination (and therefore human-associated microbiome contamination) has occurred (for an example of this and further ways to address possible contamination see Hooper R, Brealey JC, van der Valk T et al. (2018) Host-derived population genomics data provides insights into bacterial and diatom composition of the killer whale skin. Molecular Ecology). Given that some of the bacterial species identified in the study are common to human skin (e.g. Streptococcus species, which is also a common lab contaminant) I think this is a step which must be taken in order to understand whether we need to interpret the results with the caveat of contamination.

The authors must address the concern of contamination at each stage - what steps were taken to identify/remove contamination at the sampling stage (for which methodological detail must be expanded - how were samples collected? how was human contamination avoided?), at the sequencing stage (were blanks included?), and finally at the bioinformatics stage (were any bioinformatics techniques used to try to identify whether contamination had occurred?) to mitigate this potential confound.

Other concerns about the methods section are listed below:

• Tryptophan pathway: To leap from missing KOs to the whale being susceptible to colitis and CNS inflammation is extremely speculative, primarily because these KOs may be missing only post-stranding/post-death due to changes in community composition. I do not think that this analysis should be kept in the manuscript.
• Unsure what Figs 3a and 3b, and associated analysis, add to the manuscript. How do they fit with the study? What, of interest, are they showing us? The trees seem quite uninformative and the analysis doesn’t have any context in this study.
• More information is needed on the building of the cluster tree. What data did the authors use? Did they download the data from each of these species (and if so, where from?). Are they all gut microbiomes? How does this fit with the rest of the study? This is not discussed at all later – it is a very interesting result but is barely touched upon and not integrated into the manuscript. The authors should also explain why they use 16S data for this analysis, as opposed to shotgun data used in the rest of the paper (I assume so that a direct comparison between datasets can be made)
• Is gut synonymous with fecal in the methods? Confusing terminology; should be clarified.
• More details on sample collection needed. How were the samples collected?
• Line 122: do you mean adapters were removed? And why only from gut (presume fecal) and not blood/muscle?
• Typo in line 132/133: two mentions of blood and no mention of muscle
• Line 150 – 151 belongs in the discussion
• Line 165 – 166 belongs in the discussion
• Line 169: far too certain of causality

Validity of the findings

Although this study contributes novel and interesting knowledge to our understanding of the sperm whale microbiome, I have two key issues with the manuscript.

Issue 1) the interpretation of results and conclusions drawn

The authors assert that this microbiome gives clues as to why the whale stranded. The muscle and gut samples were both taken post-death of a stranded whale. After death, the microbiome is likely to change due to chemical changes in the body. For example, species of bacteria which respire anaerobically are likely to dominate (such as clostridium spp.). Thus, the microbiome seen in the recently deceased sperm whale could reflect post- mortem changes in the microbiome, as opposed to, as the author’s suggest, the microbiome of a sperm whale which is unwell and likely to strand. The blood sample was taken post-stranding, but while the whale was still alive. This is likely to show the microbiome of an individual that is stressed and, possibly, dying, but it is purely speculative to suggest that the blood microbiome of a post-stranded individual shows what the microbiome of a pre-stranded individual may look like. It is unlikely that the microbiome (blood, muscle or gut) of pre-stranding whales is likely to be similar to those of post-stranding whales given that internal conditions of the whale post-stranding are likely to differ dramatically to that of pre-stranding, and therefore different bacterial species are likely to dominate.

In brief, the authors assert that the pathogenic bacterial species (and associated virulence genes) found in this whale’s microbiome are likely to have caused its stranding. This is flawed reasoning because i) the microbiome of the stranded whale is likely to have changed post-stranding, and thus this research only sheds light on what the microbiome of a stranded sperm whale looks like, and ii) see issue 2. The manuscript therefore needs to be rephrased to take this into account. This will require a re-framing of both the introduction and the discussion.

Issue 2) pathogenic bacteria

The authors suggest that they have identified pathogenic bacterial species and parasites, which likely played some role in the death of the individual. However, many bacterial species which have been found to have pathogenic properties are actually also often found as commensal species. For example, Fusobacterium nucleatum is a commensal of the human oral microbiome, but can play a role in disease. The same is true of many Streptococcus species (which also are often found as part of the human microbiome). Whereas parasites are pathogenic, and the high quantity of them in this individual probably suggests ill health (although whether they were the cause of stranding/death is unknown as there is no healthy control whale to compare with), the bacterial species that are identified could very well be commensal species. I therefore think that the identification of possible pathogenic bacterial species should be included only in the discussion and should be highlighted as quite speculative, with a need for further study (with a healthy control for comparison).

The finding of a high abundance of parasites is interesting, but without a healthy control, it cannot be known whether this is an unusually high abundance of parasites. The authors have again made some very strong assertions that these parasites played a role in the death of the whale, but this is unfounded. At best, the authors can say that the high abundance of parasites may be related to poor pre-stranding health of the individual, however comparisons with other individuals would be needed to understand whether such high parasite loads are atypical and related in any way to the frequency of strandings.

Reviewer 2 ·

Basic reporting

The English sentence structure and grammar, word usage, and use of plurals is often not correct. For example: Line 21 ‘Mammals commensal microbiota’ should read ‘A mammal’s commensal microbiota’. The word habit is used instead of habitat, e.g. Line 45.

More discussion on the stranding of whales and the microbiome of mammals would provide a better background.

Experimental design

As this stranded whale was very near to death and the pathogens in the body were a cause it would be interested to discuss further the pathogens and the symptoms observed. This study is very opportunistic and as a result is more of a case report on a single individual.

I could not see Animal Ethics documented clearly.

Validity of the findings

The design In the materials and methods in isn’t clear what samples from which part of the animal’s body were collected. In the methods it isn’t clear how the raw sequences were treated. It is good practices to clean and trim your sequences and also to process and filter raw data such as the OTU table. I think the results you have could be discussed in the context of the current body of literature of stranding events and whale health.

Additional comments

No comment

---

## Round 0.2 · Minor Revisions

There are just a few minor edits to address by one of the reviewers prior to formal acceptance of your manuscript.

Reviewer 2 ·

Basic reporting

no comment

Experimental design

no comment

Validity of the findings

no comment

Additional comments

A survey of the sperm whale (Physeter catodon)
commensal microbiome
Many cases where plural words need to be singular or the alternative, e.g. Line 22 – plays to play; Line 72 – microbiomes to microbiome
Several lines seem to be missing words, e.g. Line 27 – that may due needs revising; Line 51 – microbes mainly needs revising
Line 49 – habit to habitat
Line 55 – has to have
Line 62 – apply these knowledge, needs revising
Line 67-68 – revise as doesn’t make sense
Many situations past and present tense are mixed, e.g. Line 71 – has not been clear, change to are not clear
Line 71 – not clear meaning not known? Please revise the wording for clarity
Line 75-77 – this sentence is better suited to the results, not the introduction
Line 96 – plural and past tense – extraction was to extractions were
Line 99 – is this Qubit 3.0 flto dissol the correct term?
Line 101 – some words seem incorrect as this sentence isn’t clear – dialed? And same?
Line 122 – use of same inn’t clear what you mean here

---

## Round 0.3 · accepted · Accept

Thank you for addressing the reviewers' comments, corrections, and suggestions.